# Changes in Attitudes toward COVID-19 Vaccination and Vaccine Uptake during Pandemic

**DOI:** 10.3390/vaccines11010147

**Published:** 2023-01-09

**Authors:** Ljiljana Markovic-Denic, Vladimir Nikolic, Nevenka Pavlovic, Gorica Maric, Aleksa Jovanovic, Aleksandra Nikolic, Vuk Marusic, Sandra Sipetic Grujicic, Tatjana Pekmezovic

**Affiliations:** 1Institute of Epidemiology, Faculty of Medicine, University of Belgrade, 11000 Belgrade, Serbia; 2Center for Disease Control and Prevention, Institute of Public Health of Belgrade, 11000 Belgrade, Serbia

**Keywords:** COVID-19, attitude, vaccination, vaccines, immunization, pandemic, follow-up

## Abstract

The epidemic control approach was based on non-pharmacological measures in the first year of the COVID-19 pandemic, followed by vaccine uptake in the second year. Vaccine uptake depends on the individual attitude toward vaccination. The aim was to assess the changes in attitudes regarding COVID-19 vaccine protection during the pandemic and to determine the vaccination uptake concerning these attitudes. A panel study on COVID-19 vaccine attitudes and vaccination against COVID-19 was conducted in Belgrade, Serbia. The first survey was carried out in May–June 2020, and the second survey was organized in August–September 2021. During the baseline testing performed in 2020, 64.4% of respondents believed that the future vaccine against COVID-19 could protect against the COVID-19 disease, while 9.7% thought that it could not, and 25.9% were unsure. One year later, in the second survey, the percentage of participants with positive attitudes was slightly lower (64.7% vs. 62.5%). However, negative attitudes turned positive in 34% of cases, and 28.9% became unsure about vaccine protection (*p* < 0.001). Out of the 390 participants included in the study, 79.7% were vaccinated against COVID-19 until follow-up. There is a statistically significant difference in vaccination uptake compared to the baseline attitude about the protection of the COVID-19 vaccine. The main finding of our study is that the majority of participants who were vaccine hesitant during the baseline testing changed their opinion during the follow-up period. Additionally, the baseline attitude about the protection of the COVID-19 vaccine has been shown to be a potential determinant of vaccination uptake.

## 1. Introduction

Following the first case of COVID-19 in Serbia on 6 March 2020, the epidemic was declared on 13 March. Thus far, 2,44 million cases and 17 thousand deaths due to COVID-19 have been reported, corresponding to a case-fatality ratio of 0.72% [1]. Initially, the epidemic control approach was based on non-pharmacological measures, including wearing masks, keeping physical distance, and lockdown. The first doses of the COVID-19 vaccine in Serbia were administered on 24 December 2020, while the mass vaccination, according to prioritized population groups, started in January 2021, just when the number of patients reached 350 thousand. Two primary doses have been required since the commencement of vaccination. A third “booster” dose was introduced in July 2021, which anticipated the recommendation given by ETAGE in November 2021 [2]. A second “booster” dose was introduced in December for people over 60 years. Already since the beginning of vaccination, five vaccines against COVID-19, based on different technologies, are available in Serbia: Sinopharm (Beijing) BBIBP-CorV (Vero Cells), BNT162b2 mRNA (Comirnaty), Pfizer–BioNTech, Sputnik V (Gam-COVID-Vac), Vaxzevri (AstraZeneca vaccine), and mRNA-1273 SARS-CoV-2 (Moderna vaccine).

Vaccine hesitancy is one of the main obstacles in bringing the COVID-19 pandemic to an end [3]. Among the different challenges in the 21st century, such as climate change, natural resources depletion, and wars, the population is faced with the COVID-19 pandemic and fear of this disease. These new priorities affect the levels of psychological well-being of the people, but also the personal attitudes towards vaccination in general, as well as immunization against COVID-19 [4].

A systematic review from 2021 has shown that since the pandemic started, the number of persons who are willing to vaccinate against COVID-19 has decreased, and the number of persons that are not willing to vaccinate has increased [5]. Besides revealing the proportion of persons that are COVID-19 vaccine-hesitant, it is crucial to detect changes in people’s attitudes regarding COVID-19 vaccination over time, i.e., to determine reasons that drive vaccine intention reversal. A growing body of knowledge on COVID-19 vaccination and factors that influence the individual decision whether to vaccinate or not could help in creating and implementing specific strategies aimed at increasing COVID-19 vaccination acceptance and, consequently, more rapid achievement of herd immunity.

Different attitudes on vaccination against COVID-19 were considered in a large number of articles. However, a much smaller number of papers have tracked the change in attitudes about vaccination. Changes in attitudes over time depend on the period when the study is conducted relative to the stage of vaccination and vary between counties and within one county. For example, two cross-sectional online studies among the working population in Hong Kong showed a decreasing willingness for COVID-19 vaccination from the first local outbreak wave in February to the third wave in September 2020 [6]. Increased concerns about vaccine safety and adequate compliance with personal protection measures explained it. In two similar cross-sectional studies during the primary and booster vaccination phases in 2021, increased vaccine hesitancy was observed in Eastern China, too [7]. However, the study conducted one year after the start of vaccination in mainland China showed a significant increase in attitudes toward the SARS-CoV-2 vaccine, which was explained by higher vaccine confidence made through tactical communication and timely disclosure of vaccine data [8]. In the USA, from early January to late March 2021, the intention to get vaccinated increased by about 18% and depended on age, race, ethnic group, and socioeconomic characteristics [8]. According to the recently published systematic review and meta-analysis of 519 articles with about 8 million participants, it was concluded that the acceptance rate declined globally in 2020, then increased in the first part of 2021, and further dropped in late 2021 [9].

Having this in mind, the aim of the present study was to assess the changes in attitudes regarding COVID-19 vaccine protection over time, from the period when vaccines were not available to about one year after vaccination started in the same cohort of the population. The secondary aim was to determine the vaccination uptake in relation to attitudes regarding the COVID-19 vaccine.

## 2. Materials and Methods

A panel study on COVID-19 vaccine attitudes and vaccination against COVID-19 was conducted within a seroepidemiology longitudinal study (EPI-COVID-SERBIA) in Belgrade, the capital of Serbia. The first survey was carried out in May–June 2020, during the end of the first wave of COVID-19 in Serbia. The second survey was organized in August–September 2021, nine months after the start of mass vaccination, during Serbia’s fifth wave of the epidemic. 

### 2.1. Study Sampling and Data Collection

A random sample of household addresses was generated at the Republic Institute of Statistics in Belgrade. The inclusion of respondents in the cohort was carried out in the period May–July 2020.

The sample size for the baseline part of our study was calculated according to the data about the seroprevalence of antibodies found in the first seroprevalence study of COVID-19 conducted in the USA, at the beginning of April 2020. After weighting for population demographics, the prevalence of antibodies at that time was estimated at 2.8% (95%CI 2.24–3.37%) [10]. According to this prevalence estimated at the time of the baseline part of the study, the reference population of Belgrade residence over 18 years of age for 2020 (1,364,599 people), margin of error of 5%, confidence interval of 95%, and assumed response rate of 50%, we calculated that a minimum of 385 included respondents would be required for an adequate analysis. To obtain this sample size, 770 respondents needed to be invited. 

The interviewer of the Institute of Public Health, an epidemiologist or specialist in another branch of preventive medicine, contacted one of the household members over 18 by phone and asked to speak with the person who is the head of the household. If he was not present, the interviewer spoke to one of the household members older than 18 years. The main goal of the research was explained to him, which is assessing the population’s immunity to COVID-19. After receiving verbal consent to participate in the study, the interviewer scheduled a visit for this family to the public health institute within five days of this telephone contact. On that occasion, the respondents signed the informed consent form for participation in the study. Then, they filled out epidemiological questionnaires, and blood was taken for serological analyses.

The self-reported questionnaire about risk perception of COVID-19 was prepared according to the questionnaire used in other counties [11]. The translation and cultural adaptation to the Serbian language was guided by the guidelines for cross-cultural adaptation of questionnaires [12]. Two forward and two back-translations were summarized and compared by the EPI-COVID-SERBIA study team. In addition to that questionnaire, several questions about vaccination attitudes were added according to the available literature [13]. The answers to those questions are included in the analysis of this survey. Then, each of the 5 researchers (GM, AJ, VN, AN, and VM) interviewed 6 respondents; thus, 30 participants aged 18 and older were interviewed in the pilot study. These questionnaires were not included in the study, but helped EPI-COVID-SERBIA to recognize possible confusion about any items and correct them. After consideration of the comprehensibility of each question, the definitive version of the questionnaire was prepared. In the validation process, we assessed the internal consistency of the questionnaire by Cronbach’s Alpha coefficient, which was 0.881.

The questionnaire contained questions about the demographic characteristics of the respondents. A question about the attitude of potential COVID-19 vaccine protection was determined as follows: “COVID-19 vaccine can help protect against disease”. Furthermore, one question was related to the respondent’s opinion of “can get COVID-19 if he would not be vaccinated when the vaccine will be available” Possible responses for both questions were “yes”, “no”, and “unsure”. 

All respondents of this first survey were invited by the same interviewer to participate in the follow-up, second survey during the period August–September 2021. Besides serologic testing, they filled out questions about attitudes toward vaccines if they were vaccinated. If they were vaccinated, they stated which vaccine they received and whether they received the complete first series of two vaccines. If they were not vaccinated, they indicated their reasons for not being vaccinated. Moreover, they stated their opinion about the protection (vaccine effectiveness) of the vaccine, that is, whether the vaccine can prevent the onset of illness and death.

### 2.2. Statistical Analysis

In data processing, we used descriptive and analytical statistical methods. Data are presented as mean ± SD for continuous variables and number (percentage) for categorical variables. The sample size for the baseline part of our study was calculated by a sample size calculator, based on the reference population that was estimated for 2020 by the Statistical Office of the Republic of Serbia (https://www.stat.gov.rs/en-US accessed on 16th April 2020). 

A chi-squared test was used to analyze categorical data between compared groups, and an independent t-test was used for continuous variables. For analysis of baseline and follow-up attitude on COVID-19 vaccine protection against the disease, which represented repeated measures, the McNemar–Bowker test was performed. Statistical analysis was performed using SPSS version 26.0 software (SPSS Inc., Chicago, IL, USA).

## 3. Results

Out of 693 participants tested during the first survey, 390 (56.3%) completed the follow-up survey one year later. The demographic characteristics of participants in both surveys concerning the COVID-19 vaccine protection attitude are shown in Table 1. There were more female respondents (59.7%). The mean age of participants was 51.4 ± 17.0 years. There was no statistically significant difference regarding the baseline attitude according to the gender and age of the respondents. More participants with lower education were unsure whether the COVID-19 vaccine could protect against the disease, although the difference was not statistically significant. The seroprevalence during baseline testing was 8.3% (95%CI: 5.7–11.5), while at the follow-up, it was 57.5% (95%CI: 52.4–62.5).

During the baseline testing performed in 2020, 64.4% (250/390) of respondents believed that the future vaccine against COVID-19 could protect against the COVID-19 disease, while 9.7% (38/390) thought that it could not, and 25.9% (101/390) were unsure.

At the follow-up, 62.5% (232/371) of the participants believed that the vaccine against COVID-19 can protect against the COVID-19 disease, 17.8% believed that it does not, and 19.7% were unsure. As many as 62.9% of respondents who had a negative attitude or were unsure at the baseline regarding the future COVID-19 vaccine’s ability to protect changed their opinion during the follow-up. On the contrary, the largest number of respondents who stated at the baseline testing that the future COVID-19 vaccine could protect against the disease maintained their attitude at the follow-up (Table 2, Figure 1).

Most respondents at the follow-up (81.8%) believed that the COVID-19 vaccine could protect against serious illness or death. However, of the participants that stated on the baseline that the future COVID-19 vaccine would not protect against the disease, at the follow-up, 70.6% stated that the vaccine could protect against severe illness or death, while only 11.8% stated it could not (*p* < 0.001) (Table 2).

Out of the 390 participants included in the study, 311 (79.7%) were vaccinated against COVID-19 until follow-up. There is a statistically significant difference in vaccination uptake compared to the baseline attitude about the protection of the COVID-19 vaccine (Table 3). The largest number of vaccinated participants received 2 doses, while in the follow-up survey, only 6.4% of respondents were vaccinated with 3 doses of the vaccine. Most often, respondents were vaccinated with the Sinopharm (BBIBP-CorV) vaccine, followed by Pfizer-BioNTexh and Sputnik V (Gam-COVID-Vac), and the smallest number of respondents were vaccinated with AstraZeneca (Vaxzevria). The reasons participants stated for not being vaccinated were that the vaccine was not sufficiently tested (27.5%), they did not believe in vaccines (22.5%), insufficient information (22.5%), fear of a possible adverse event (13.8%), underlying disease (5.0%), and other reasons (20%) (Table 3).

During the baseline testing, 26.4% of the participants believed there was no chance that they could get COVID-19 if they did not receive the vaccine, while the same number stated that they could get COVID-19 if they did not get vaccinated, and 47.1% were unsure. Vaccination characteristics of participants based on attitude toward the possibility of getting COVID-19 are presented in Table 4.

## 4. Discussion

Two consecutive surveys were conducted before and after the vaccine became available in Serbia to investigate the changes in attitudes toward COVID-19 vaccines and vaccination. Our survey showed that about two-thirds of respondents had a positive attitude about vaccine protection against COVID-19 before vaccines were produced. A similar study was conducted in China at the end of 2020, one month before the COVID-19 vaccine became available [14]. In this study, approximately half of the respondents expressed no hesitancy toward vaccines, while 28% were hesitant, and 21% refused to be vaccinated if the vaccine would be developed and approved [14]. Keeping in mind that China has its own vaccine production, our finding was encouraging at that moment. Indeed, about a fifth of the population completed the primary COVID-19 protocol during the first three months since the vaccination started. In a large population survey in Australia, when vaccination had just started in this country, and only 2% of the population were vaccinated, 78% of individuals reported that they were likely to get the SARS-CoV-2 vaccine. About 15% of participants were unsure about vaccination [15]. An online survey among representative samples of the population in seven countries in western Europe was conducted in April 2020 during the clinical and preclinical evaluation phases of COVID-19 vaccines. At that time, 74% of the participations stated their willingness to vaccinate against COVID-19 when the COVID-19 vaccine would be available [16]. Data collected in the online cross-sectional study in European countries just before Europe’s COVID-19 vaccine rollout showed that 57% of respondents would accept a COVID-19 vaccine, 19.0% would not, and 24.1% were unsure [17]. In the international cross-sectional iCARE study conducted over three periods, four European counties participated among eight countries. The first time period lasted from the end of March to the end of May 2020, similar to our first survey. In all countries, vaccine hesitancy was 27%; 25.6% in the first period. The responders in France showed the highest rate of vaccine hesitancy, from 39.9% in the first period to 51% in the third period, which ended at the end of January 2021, before mass vaccination started. In Italy, the lowest rate of hesitancy was reported at the beginning of the pandemic (9.3%), but hesitancy increased up to 19% by the end of the survey [18]. A study in that country revealed that more people infected with COVID-19, even with mild disease symptoms, showed higher vaccine hesitancy than those who did not develop COVID-19 [19]. Individuals who had experienced economic stress and those with a negative opinion about the government’s response were less likely to accept immunization in Sweden [20]. The trust in non-pharmaceutical interventions, confidence in institutions, and demographic factors were associated with willingness for COVID-19 vaccination in Germany [21]. All the studies above were related to non-European or Western European countries. One of the few articles includes a literature review during 2020–2021 from Eastern and Southern European countries. Out of 223 studies, 44 cross-sectional studies were included in the final evaluation concerning confidence in vaccines’ safety and efficacy; confidence in the healthcare system, government, and public health measures; trust in the healthcare system; and government and public health measures. This review showed that individual perceptions play a significant role in the decision to vaccinate against COVID-19 [22].

In our first survey, 25.9% of participants were unsure about the vaccine’s protective effect, while 9.7% had an entirely negative opinion about vaccine protection. 

In a large number of studies, it has been shown that vaccine hesitancy decreased with higher education level [23,24]. A lower level of education is associated with a lower vaccination rate against COVID-19 [25,26]. In the systematic review and meta-analysis of COVID-19 vaccination willingness, out of 39 studies, 25 included education as a predictor for vaccination. A borderline significant association between education level and vaccination willingness was found in 14 studies. In the study conducted in eight EU counties, the level of education did not correlate with vaccine hesitancy, while in five countries, vaccine hesitancy was associated with lower education [27]. Although the largest percentage of our respondents who were unsure about the role of the vaccine in the prevention of COVID-19 had lower education, the statistical difference was not significant.

One year later, in the second survey, slightly less than two-thirds of participants who did not believe that vaccines can protect against the disease changed their opinion. Furthermore, only 37% maintained a negative attitude about vaccine protection. Although there were no targeted communication programs with the population in our country that would raise citizens’ awareness about the importance of immunization in disease prevention, vaccination against COVID-19 was promoted in the media. Eminent experts made guest appearances on various TV channels, explaining the importance of immunization. Moreover, there was a lot of promotional material about COVID-19 vaccines on the websites of official organizations and institutions in our country. The campaign called “Let’s return the hug,” supported by the Government and Ministry of Health, was implemented by all public health institutes and primary healthcare centers. In the study conducted in Hong Kong, it has been stated that respondents reported having vaccine hesitancy may not necessarily translate to no vaccine uptake, which was a similar finding to our results [28].

All of the above contributed to an increased awareness of the importance of all preventive measures, especially vaccination against COVID-19. It is well known that social media’s role in public health interventions is to mitigate and reduce vaccine hesitancy [29]. However, misinformation about COVID-19 vaccines can spread on social media platforms and networks [30,31], sometimes leading to the so-called infodemic. An infodemic is defined as “too much information, including false or misleading information in digital and physical environments during a disease outbreak.” [32].

It should be pointed out that, in Serbia, there is a well-established tradition of vaccinating children in pediatric departments of primary healthcare centers and adults in 25 public health institutes, including the National Institute of Public Health of Serbia. A lot of written material, pamphlets, and leaflets were prepared in these centers and distributed to parents when they brought their children for examinations and to citizens who came for preventive examinations.

One of the possible reasons for the change in attitudes toward vaccination is the availability of several types of vaccines in Serbia. Citizens are able to decide for themselves which type of COVID-19 vaccine they wanted to receive. However, doctors were present at all vaccination points. In addition to the mandatory pre-vaccination examination, they explained the type of vaccine and directly communicated the information so that a broad audience could understand it. It is important to emphasize that in Serbia, vaccination for children is carried out exclusively in pediatric departments of primary healthcare centers. In addition to the vaccination in the primary healthcare centers, special COVID-19 vaccination points were set up for adults. Therefore, citizens could come to the vaccination point just for information and get vaccinated on another occasion.

The largest percentage of respondents received two doses of the vaccine from the primary series, because the booster dose is recommended in Serbia immediately before the second survey. The majority of our subjects decided to receive the Sinopharm vaccine, but the difference in the choice for a certain type of vaccine did not depend on the attitude towards vaccines in the baseline survey. The Pfizer and Sinopharm vaccines were the first that individuals received in Serbia. During the very short period of time after the start of mass vaccination in January 2021, the other three vaccine types were also delivered in our country. Few countries in Europe have been able to offer their citizens a choice of one of several available vaccines based on different manufacturing technology. For example, five types of COVID-19 vaccines were offered to the population in Hungary [33]. Usually, two or three COVID-19 vaccines are available in many countries [34,35].

On the one hand, the availability of several vaccines can lead to uncertainty about which vaccine to choose. On the other hand, it is an advantage, because free vaccine choice reduces hesitation regarding vaccination and increases its acceptance [36].

Until January 2022, the total COVID-19 vaccination rate with two vaccines in the primary series varied from 33% to 72% in the Easter European countries, while the mean rate in the EU countries was 74% [18]. After the exceptional success in vaccination coverage in 2021, declining interest in vaccination of the population of Serbia was further observed, resulting in 47.7% vaccination coverage in mid-2022 [37].

In addition to personal attitudes towards vaccination, external factors, such as military conflict, sanctions, and economic crises, can lead to vaccine shortages, reduce coverage, and change attitudes about vaccination in the long term, which was observed in other countries and in Serbia, as well [38,39,40,41,42].

The majority of non-vaccinated persons justified their hesitancy toward vaccination by the belief that the new vaccines had not been tested enough or that they did not have enough information about vaccines or that they did not believe in the effectiveness of vaccines. The study of vaccine hesitancy in eight European countries revealed that receiving messages about medical and hedonistic benefits of vaccination can increase vaccine coverage [28]. This argues for a good practice of providing comprehensive information about vaccines at vaccination points in our country.

In the large international study, authors investigated variables related to having a positive attitude as factors that could potentially increase the uptake of vaccines. They found that the following variables could improve people’s attitudes towards vaccines: better information about the COVID-19 vaccines in general, and especially information related to the safety and side effects of vaccines; increased trust in governments in conducting the vaccine rollout; and handling procurement and capacity issues [43].

Strengths of our study include the representative sample and the fact that similar studies had not been performed earlier in the Serbian population, while there are only a few studies in other populations. Namely, previous studies mostly used a cross-sectional study design, while our study observed same participants at two different time points during the COVID-19 pandemic. One of the limitations of this study is that it was only conducted on a random sample of the population of Belgrade, the capital city, and cannot be representative of the whole population of Serbia. It is very likely that the availability of information through various social media is more significant in the largest city than in smaller towns in the country. The level of education and biological literacy affect the awareness of the disease and the attitude about the vaccine’s protective importance, which was revealed in other studies [27]. However, the availability of the vaccine and the organization of vaccine uptake were the same in all places in Serbia. It is well-known that one of the limitations of the study can be a low participation rate (PR), namely, the proportion of members who refuse to participate in the study. We observed that PR in our baseline study was 42.9%. In the recently published article that analyzed PR from 90 seroprevalence studies, the calculated rate from 35 definitely included studies varied from 0.43 to 96.4, with a mean of 63% [44]. In order to increase the PR in the study, our interviewers gave every potential participant who was undecided whether to participate in the study the opportunity to make a definitive decision, with a prior explanation of the importance of the study, both on the public health level and on the physical level, i.e., obtaining antibody level results when it could not be done simply in a routine procedure in the health service. Each undecided person was invited once more at the scheduled time. Another limitation could refer to information bias because we used self-reported data obtained by means of questionnaire.

## 5. Conclusions

The main finding of our study is that the majority of participants who were vaccine hesitant during the baseline testing changed their opinion during the follow-up period. Additionally, the baseline attitude about the protection of the COVID-19 vaccine has been shown to be a potential determinant of vaccination uptake. Spontaneous promotion of vaccination, together with other public health interventions in our country, potentially contributed to changes in attitudes regarding COVID-19 immunization. From a scientific point of view, the concept of vaccination promotion should be extended to building a communication strategy in the population, where the main principles would be transparency, consistency, predictability, and involving the public to become part of the solution. Such strategies could potentially impact global public health in COVID-19 pandemic conditions.

## Figures and Tables

**Figure 1 vaccines-11-00147-f001:**
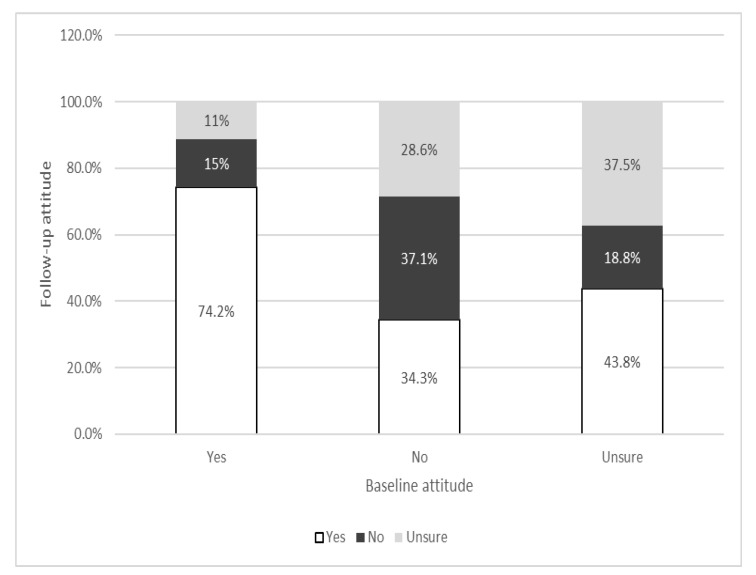
Changes in attitudes toward COVID-19 vaccine protection: baseline and follow-up attitudes (“COVID-19 vaccine can protect against the disease”).

**Table 1 vaccines-11-00147-t001:** Demographic characteristics of participants based on COVID-19 vaccine protection attitude.

	Baseline Attitude: COVID-19 Vaccine Can Protect against the Disease				
	Yes n (%)	No n (%)	Unsure n (%)	Total n (%)	*p* Value
Total	251 (64.4)	38 (9.7)	101 (25.9)	390 (100.0)	
Gender					
Male	108 (43.0)	15 (39.5)	34 (33.7)	157 (40.3)	0.268 *
Female	143 (57.0)	23 (60.5)	67 (66.3)	233 (59.7)	
Age (years)—mean ± SD	52.4 ± 16.9	50.5 ± 16.2	48.9 ± 17.2	51.4 ± 17.0	0.195 **
18–20 years	9 (3.6)	2 (5.3)	3 (3.0)	14 (3.6)	0.106 *
21–30 years	22 (8.8)	3 (7.9)	15 (14.9)	40 (10.9)	
31–40 years	36 (14.3)	5 (13.2)	18 (17.8)	59 (15.1)	
41–50 years	43 (17.1)	6 (15.8)	20 (19.8)	69 (17.7)	
51–60 years	37 (14.7)	12 (31.6)	7 (6.9)	56 (14.4)	
61–65 years	35 (13.9)	3 (7.9)	11 (10.9)	49 (12.6)	
>65 years	69 (27.5)	7 (18.4)	27 (26.7)	104 (26.6)	
Education					
Primary	15 (6.1)	2 (5.4)	7 (7.1)	24 (6.3)	0.815 *
Secondary	119 (48.6)	18 (48.6)	52 (52.5)	189 (49.6)	
College	11 (4.5)	1 (2.7)	3 (3.0)	15 (3.9)	
Bachelor	97 (39.6)	15 (40.5)	33 (33.3)	145 (38.1)	
Masters	3 (1.2)	1 (2.7)	4 (4.0)	8 (2.1)	
Occupation					
Unemployed	5 (2.0)	1 (2.7)	1 (1.0)	7 (1.8)	0.657 *
Student	10 (4.1)	5 (13.5)	11 (11.1)	26 (6.8)	
Technicians and associate professionals	19 (7.8)	1 (2.7)	8 (8.1)	28 (7.3)	
Service and sales workers	20 (8.2)	2 (5.4)	3 (3.0)	25 (6.6)	
Clerical support workers	14 (5.7)	1 (2.7)	3 (3.0)	18 (4.7)	
Legal, social and cultural professionals	35 (14.3)	5 (13.5)	10 (10.1)	50 (13.1)	
Teaching professionals	13 (5.3)	2 (5.4)	6 (6.1)	21 (5.5)	
Managers, Information and communications technology professionals	8 (8.1)	2 (5.4)	5 (5.1)	15 (6.6)	
Health professionals	17 (6.9)	3 (8.1)	5 (5.1)	25 (6.6)	
Science and engineering professionals	23 (9.4)	5 (13.5)	9 (9.1)	37 (9.7)	
Retired	74 (30.2)	10 (27.0)	35 (35.4)	119 (31.2)	
Other	7 (2.9)	0 (0.0)	3 (3.0)	10 (2.6)	

* Chi-squared test; ** Independent t-test.

**Table 2 vaccines-11-00147-t002:** Change of attitude regarding COVID-19 vaccine protection.

Follow-Up Attitude:	Baseline Attitude: COVID-19 Vaccine Can Protect against the Disease				*p* Value
	Yes n (%)	No n (%)	Unsure n (%)	Total n (%)	
**COVID-19 Vaccine Can Protect Against the Disease**					**0.001 ***
Yes	178 (74.2)	12 (34.3)	42 (43.8)	232 (62.5)	
No	35 (14.6)	13 (37.1)	18 (18.8)	66 (17.8)	
Unsure Total	27 (11.3) 240 (64.7)	10 (28.6) 35 (9.4)	36 (37.5) 96 (25.9)	73 (19.7) 371 (100.0)	
**COVID-19 vaccine can protect against** **severe disease or death**					
Yes	218 (89.0)	24 (70.6)	67 (67.7)	310 (81.8)	**<0.001 ****
No	9 (3.7)	4 (11.8)	8 (8.1)	21 (5.5)	
Unsure Total	18 (7.3) 245 (64.6)	6 (17.6) 34 (9.0)	24 (24.2) 99 (26.1)	48 (12.7) 379 (100.0)	

* McNemar–Bowker test; ** Chi-squared test.

**Table 3 vaccines-11-00147-t003:** Vaccination characteristics of participants based on baseline COVID-19 vaccine protection attitude.

	Baseline Attitude: COVID-19 Vaccine Can Protect against the Disease				*p* Value *
Vaccinated Against COVID-19	Yes n (%)	No n (%)	Unsure n (%)	Total n (%)	
Yes	224 (89.2)	23 (60.5)	64 (63.4)	311 (79.7)	**<0.001**
No	27 (10.8)	15 (39.5)	37 (36.6)	79 (20.3)	
**Doses**					
One dose	5 (2.2)	0 (0.0)	4 (6.3)	9 (2.9)	**<0.001**
Two doses	199 (89.2)	22 (95.6)	56 (87.3)	277 (89.1)	
Three doses	20 (9.0)	1 (4.4)	4 (6.3)	25 (8.0)	
**Vaccine manufacturer**					
Pfizer-BioNTexh	41 (19.1)	6 (26.1)	12 (18.8)	59 (19.5)	0.704
Sinopharm (BBIBP-CorV)	135 (62.8)	12 (52.2)	44 (68.8)	192 (63.4)	
Sputnik V (Gam-COVID-Vac)	22 (10.2)	4 (17.4)	5 (7.8)	31 (10.2)	
AstraZeneca (Vaxzevria)	17 (7.9)	1 (4.3)	3 (4.7)	21 (6.9)	
**Reasons for not being vaccinated**					
Underlying disease	1 (3.7)	2 (13.3)	1 (2.6)	4 (5.0)	0.254
Insufficient information	5 (18.5)	2 (13.3)	11 (28.9)	18 (22.5)	0.392
I don’t believe in vaccines	8 (29.6)	1 (6.7)	9 (23.7)	18 (22.5)	0.226
Possible adverse event	2 (7.4)	3 (20.0)	6 (15.8)	11 (13.8)	0.462
The disease is not serious	3 (11.1)	0 (0.0)	0 (0.0)	3 (3.8)	0.047
The vaccine has not been sufficiently tested	5 (18.5)	5 (33.3)	12 (31.6)	22 (27.5)	0.435
Other	7 (25.9)	2 (13.3)	7 (18.4)	16 (20.0)	0.586

* Chi-squared test.

**Table 4 vaccines-11-00147-t004:** Vaccination characteristics of participants based on attitude toward the possibility of getting COVID-19.

	Baseline Attitude: I Can Get COVID-19 if I Don’t Get Vaccinated				*p* Value *
Vaccinated against COVID-19	Yes n (%)	No n (%)	Unsure n (%)	Total n (%)	
Yes	93 (92.1)	64 (64.0)	145 (81.0)	302 (79.5)	**<0.001**
No	8 (7.9)	36 (36)	34 (19.0)	78 (20.5)	
**Doses**					
One dose	4 (4.0)	2 (2.0)	3 (1.7)	9 (2.4)	**<0.001**
Two doses	77 (76.2)	59 (58.4)	133 (74.3)	269 (70.6)	
Three doses	12 (11.9)	4 (4.0)	9 (5.0)	25 (6.6)	
**Vaccine manufacturer**					
Pfizer	16 (18.0)	14 (22.6)	29 (20.1)	59 (20.0)	0.807
Sinopharm	62 (69.7)	37 (59.7)	90 (62.5)	189 (64.1)	
Sputnik V	7 (7.9)	5 (8.1)	15 (10.4)	27 (9.2)	
AstraZeneca	4 (4.5)	6 (9.7)	10 (6.9)	20 (6.8)	

* Chi-squared test.

## Data Availability

Data available on request due to restrictions. The data presented in this study are available on request from the corresponding author. The data are not publicly available due to privacy protection.

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
