# Peer review of "Changes in Attitudes toward COVID-19 Vaccination and Vaccine Uptake during Pandemic"

_vaccines, 2023, doi:10.3390/vaccines11010147_

Round 1
Reviewer 1 Report
It is a very comprehensive manuscript and well done research in order to understand the position of people regarding their vaccination position.
Author Response
Comments from Reviewer 1
It is a very comprehensive manuscript and well done research in order to understand the position of people regarding their vaccination position.
Response: We want to thank the Reviewer very much for this kind comment.
Reviewer 2 Report
The article is devoted to assessing the change in attitudes towards vaccine protection against COVID-19 during a pandemic and determining the use of vaccination in connection with this attitude. The approach to combating the epidemic justified the study's relevance and was based on non-drug measures in the first year of the COVID-19 pandemic, followed by using a vaccine in the second year. At the same time, the perception of the vaccine depends on the individual attitude towards vaccination. Therefore, this study aimed to evaluate changes in attitudes towards vaccine protection against COVID-19 during the pandemic and to determine the use of vaccination concerning this attitude. A panel study of attitudes towards the COVID-19 vaccine and COVID-19 vaccination was conducted in Belgrade, Serbia. The first survey was conducted in May-June 2020, and the second was organized in August-September 2021. In baseline testing conducted in 2020, 64.4% of respondents believed that a future COVID-19 vaccine could protect against COVID-19. -19, with 9.7% believing tcouldn'tdn't and 25.9% not being sure. A year later, in the second survey, the percentage of participants with a positive attitude was somewhat lower (64.7% vs. 62.5%). However, negative attitudes changed to positive in 34% of cases, and 28.9% became uncertain about vaccine protection (p<0.001). Of the 390 participants included in the study, 79.7% were vaccinated against COVID-19 before follow-up.
Despite the satisfactory quality of the article, some shortcomings need to be corrected.
- It is recommended to include the Current research analysis section to review the existing approaches of attitude toward vaccination estimation briefly.
- It is recommended to expand subsection 2.2. with a detailed description of methods used within the study.
- The visualization of the results will increase the quality of the paper.
- It is recommended to include the discussion of external factors influencing the vaccination and attitude toward it, e.g. doi: 10.1136/bmjgh-2022-009173
- The scientific and practical novelty of the research should be highlighted in the Conclusions section.
- Please, check the correctness of sections “Funding”, “Institutional Review Board Statement”, “Data Availability Statement”.
In summarizing my comments, I recommend that the manuscript is accepted after minor revision.
Author Response
Comments from Reviewer 2
The article is devoted to assessing the change in attitudes towards vaccine protection against COVID-19 during a pandemic and determining the use of vaccination in connection with this attitude. The approach to combating the epidemic justified the study's relevance and was based on non-drug measures in the first year of the COVID-19 pandemic, followed by using a vaccine in the second year. At the same time, the perception of the vaccine depends on the individual attitude towards vaccination. Therefore, this study aimed to evaluate changes in attitudes towards vaccine protection against COVID-19 during the pandemic and to determine the use of vaccination concerning this attitude. A panel study of attitudes towards the COVID-19 vaccine and COVID-19 vaccination was conducted in Belgrade, Serbia. The first survey was conducted in May-June 2020, and the second was organized in August-September 2021. In baseline testing conducted in 2020, 64.4% of respondents believed that a future COVID-19 vaccine could protect against COVID-19. -19, with 9.7% believing tcouldn'tdn't and 25.9% not being sure. A year later, in the second survey, the percentage of participants with a positive attitude was somewhat lower (64.7% vs. 62.5%). However, negative attitudes changed to positive in 34% of cases, and 28.9% became uncertain about vaccine protection (p<0.001). Of the 390 participants included in the study, 79.7% were vaccinated against COVID-19 before follow-up.
Despite the satisfactory quality of the article, some shortcomings need to be corrected.
- It is recommended to include the Current research analysis section to review the existing approaches of attitude toward vaccination estimation briefly.
Response: Thank you very much for this constructive input. We added one paragraph in the Introduction with a brief review of recently published papers on attitude changes and acceptance of vaccination against COVID-19.
- It is recommended to expand subsection 2.2. with a detailed description of methods used within the study.
Response: We thank the Reviewer for this valuable suggestion. Subsection 2.2 was expanded accordingly. We added one paragraph about sample size. We also added one paragraph about the participation rate in the seroprevalence studies, in the Discussion, section about the limitation of the study.
The visualization of the results will increase the quality of the paper.
Response: Thank you very much for this significant insight. We have now added Figure 1 in order to make results clearer and to facilitate their interpretation.
- It is recommended to include the discussion of external factors influencing the vaccination and attitude toward it, e.g. doi: 10.1136/bmjgh-2022-009173
Response: We thank the Reviewer for this important suggestion. We have now cited this reference (now, No 27) and added references 28-31, following text to Discussion section: “In addition to personal attitudes towards vaccination, some external factors such as military conflict, sanctions, and economic crises can lead to vaccine shortages, reduce coverage, and change attitudes about vaccination in the long term, which was observed in other countries and in our county as well”
- The scientific and practical novelty of the research should be highlighted in the Conclusions section.
Response: Thank you for this significant comment. In addition to the practical novelty that was already highlighted in the Conclusion, as “a negative attitude towards vaccine acceptance can be changed by well-designed health promotion strategies and vaccination campaigns with the support of the whole society”, we have now emphasized the scientific novelties: “From a scientific point of view, the concept of vaccination promotion should be extended to building a communication strategy in the population, where the main principles would be transparency, consistency, predictability, and involving the public to become part of the solution.
- Please, check the correctness of sections “Funding”, “Institutional Review Board Statement”, “Data Availability Statement”.
Response: Thank you very much. We have performed double-check of our manuscript and edited these subsections.
In summarizing my comments, I recommend that the manuscript is accepted after minor revision.
Response: We want to thank the Reviewer for all the valuable comments and suggestions that improved our manuscript.
Reviewer 3 Report
I was invited to revise the paper entitled "Changes of attitudes toward COVID-19 vaccination: two-periods panel data analysis". It reports results from a survey performed among Serbian inhabitants aimed to evaluate attitudes and adherence toward covid vaccination. After, the same sample was invited to partecipate to a second survey one year after the first one.
The topic is very interesting and poor study were conducted in Serbia on this topic.
Major observations:
- Sample size estimation was lacking;
- One of the main problem of this paper was the lack of information about education level, job and economic income: these factors are one of the most important associated factor to vaccine hesitancy;
- Among discussions, strenght and limitation section was lacking;
- Results of blood sampling were not presented.
Minor observations:
- Englsh language whould be revised (example in ttitle: "Change in" instead of "Change of";
- Astrazeneka should be corrected with "Astrazeneca";
- Authors should compare their results with similar study performed in other countries.
Author Response
Comments from Reviewer 3
I was invited to revise the paper entitled "Changes of attitudes toward COVID-19 vaccination: two-periods panel data analysis". It reports results from a survey performed among Serbian inhabitants aimed to evaluate attitudes and adherence toward covid vaccination. After, the same sample was invited to participate to a second survey one year after the first one.
The topic is very interesting and poor study were conducted in Serbia on this topic.
Response: We thank the Reviewer for this kind comment.
Major observations:
- Sample size estimation was lacking;
Response: We thank the Reviewer for this observation. We added one paragraph about sample size in section 2.2 of Method. We also added one paragraph about the participation rate in the seroprevalence studies, in the Discussion, section about the limitation of the study.
- One of the main problem of this paper was the lack of information about education level, job and economic income: these factors are one of the most important associated factor to vaccine hesitancy;
Response: Thank you very much for this constructive comment. We did an additional analysis and added a comparison of the vaccine protection attitude, according to the level of education (primary, secondary, college, bachelor, and masters) and according to the occupation (Table 1). Unfortunately, we do not have data about the economic income of our participants. Besides, we added one paragraph concerning education and occupation in the Discussion.
- Among discussions, strength and limitation section was lacking;
Response: We thank the reviewer for the suggestion to further expand on this issue. We have added the two paragraphs in Discussion section, strengths, and limitations.
- Results of blood sampling were not presented.
Response: Although the results of blood testing were not the subject of this manuscript, we added the seroprevalence values during baseline testing and at follow-up, with 95%CI in the Results section, the last sentence of the first paragraph, as follows: “The seroprevalence during baseline testing was 8.3% (95%CI 5.7-11.5), while at the follow-up it was 57.5% (95%CI 52.4-62.5 )”
Round 2
Reviewer 3 Report
Authors addressed the great part of my previous comments. I suggest to improve english languages and to improve discussion section comparing their results with similar studies performed across Europe.
Author Response
Reviewer 3#:
( ) I would not like to sign my review report
(x) I would like to sign my review report
English language and style
( ) English very difficult to understand/incomprehensible
( ) Extensive editing of English language and style required
(x) Moderate English changes required
( ) English language and style are fine/minor spell check required
( ) I don't feel qualified to judge about the English language and style
Comments and Suggestions for Authors
Authors addressed the great part of my previous comments.
I suggest to improve english languages and to improve discussion section comparing their results with similar studies performed across Europe.
Response: Thank you for this suggestion. Our manuscript is checked by a
native English-speaking colleague.
We added eight references (new No 13, 14, 16, 17, 18, 19) about studies conducted in Europe and compared our results with the results from these. We also added references No 25 and 34.